# Data-driven estimation of the instantaneous reproduction number and growth rates for the 2022 monkeypox outbreak in Europe

**Fernando Saldaña**[1]*, **Maria L. Daza-Torres**[2], **Maíra Aguiar**[1,3,4]

**1** Basque Center for Applied Mathematics (BCAM), Bilbao, Spain, **2** Department of Public Health Sciences, University of California Davis, Davis, California, United States of America, **3** Ikerbasque, Basque Foundation for Science, Bilbao, Spain, **4** Dipartimento di Matematica, Università degli Studi di Trento, Trento, Italy

* fsaldana@bcamath.org

**Data Availability Statement:** The datasets analyzed during the current study are available from an open-access database presented in Kraemer, M. U., Tegally, H., Pigott, D. M.,

## Abstract

### Objective

To estimate the instantaneous reproduction number $R_t$ and the epidemic growth rates for the 2022 monkeypox outbreaks in the European region.

### Methods

We gathered daily laboratory-confirmed monkeypox cases in the most affected European countries from the beginning of the outbreak to September 23, 2022. A data-driven estimation of the instantaneous reproduction number is obtained using a novel filtering type Bayesian inference. A phenomenological growth model coupled with a Bayesian sequential approach to update forecasts over time is used to obtain time-dependent growth rates in several countries.

### Results

The instantaneous reproduction number $R_t$ for the laboratory-confirmed monkeypox cases in Spain, France, Germany, the UK, the Netherlands, Portugal, and Italy. At the early phase of the outbreak, our estimation for $R_t$, which can be used as a proxy for the basic reproduction number $R_0$, was 2.06 (95% CI 1.63 − 2.54) for Spain, 2.62 (95% CI 2.23 − 3.17) for France, 2.81 (95% CI 2.51 − 3.09) for Germany, 1.82 (95% CI 1.52 − 2.18) for the UK, 2.84 (95% CI 2.07 − 3.91) for the Netherlands, 1.13 (95% CI 0.99 − 1.32) for Portugal, 3.06 (95% CI 2.48 − 3.62) for Italy. Cumulative cases for these countries present subexponential rather than exponential growth dynamics.

### Conclusions

Our findings suggest that the current monkeypox outbreaks present limited transmission chains of human-to-human secondary infection so the possibility of a huge pandemic is very low. Confirmed monkeypox cases are decreasing significantly in the European region, the decline might be attributed to public health interventions and behavioral changes in the

Dasgupta, A., Sheldon, J., Wilkinson, E.,. . . & Brownstein, J. S. (2022). Tracking the 2022 monkeypox outbreak with epidemiological data in real-time. The Lancet Infectious Diseases. https://doi.org/10.1016/S1473-3099(22)00359-0.

**Funding:** FS and MA are supported by the Basque Government through the "Mathematical Modeling Applied to Health" BMTF Project, BERC 2022-2025 program and by the Spanish Ministry of Sciences, Innovation, and Universities: BCAM Severo Ochoa accreditation CEX2021-001142-S / MICIN / AEI / 10.13039/501100011033. MLDT received no specific funding for this work. The funders had no role in study design, data collection and analysis, decision to publish, or preparation of the manuscript. There was no additional external funding received for this study.

**Competing interests:** The authors have declared that no competing interests exist.

population due to increased risk perception. Nevertheless, further strategies toward elimination are essential to avoid the subsequent evolution of the monkeypox virus that can result in new outbreaks.

## Introduction

The monkeypox virus is an enveloped double-stranded DNA virus discovered in a Danish laboratory in 1958 and is known for being the etiological agent of a zoonotic disease known as monkeypox (mpox) whose first human cases were identified in the Democratic Republic of Congo in 1970. Considering genetic and geographic variation, there are two distinct clades of the virus: the central African (also known as Congo Basin) and the west African clade, the former being the more virulent based on a higher case fatality ratio [1]. Since 1970, mpox has remained endemic in central and west Africa, with only a few cases outside of this region. Yet, in 2003 an outbreak was documented in the United States of America after the importation of infected animals. Until May 2022, all laboratory-confirmed mpox cases identified outside of West and Central Africa were either imported or linked to an imported case or animals imported from endemic areas [1]. On 16 May 2022, the United Kingdom Health Security Agency reported the identification of four cases of mpox with no history of recent travel to endemic areas or contact with previously reported cases. Afterward, multiple outbreaks were reported in several countries. Hence, on 23 July, the WHO declared the outbreak a public health emergency of international concern [2]. The outbreak has now affected more than 100 countries and more than 80, 000 cases have been confirmed as of 14 November 2022. Europe was the epicenter of this large and geographically widespread outbreak approximately until the end of July 2022. Later, the region of the Americas became the area with the highest number of confirmed cases [2].

People can not only acquire the mpox virus after being in contact with an infected human but also via a spillover event after direct contact with body fluids, skin, or mucosal lesions of infected wildlife. Rodents, squirrels, non-human primates, and other species can be infected, but the natural reservoir has not yet been confirmed. Human-to-human transmission occurs mainly through close contact with infectious material from skin lesions of an infected person and respiratory droplets after prolonged close contact [1]. The present outbreak has affected predominantly the gay and bisexual men who have sex with men (GBMSM) community, with the nature of the presenting lesions suggesting that transmission occurred during sexual intercourse, although some cases have also been detected in women and children [2]. Mass gatherings events such as summer music festivals and specific sexual practices have facilitated the transmission of mpox among the GBMSM group [3]. However, at present, there is not enough evidence to confirm that mpox transmission occurs mainly via sexual routes [4]. Furthermore, while detecting and preventing human-to-human secondary infections are critical, transmission from animals to humans is also a major concern. If the virus becomes endemic in an animal reservoir outside of endemic areas, it would mean a continuous risk of repeated human outbreaks, and disease elimination will not be reachable [1].

As the mpox outbreak unfolds, mathematical modeling can be used as a public health guidance tool to evaluate the impact of control interventions and better understand the epidemiology of the outbreak [5]. Real-time monitoring of changes in transmissibility can be done by virtue of the instantaneous reproduction number $R_t$ (also called effective reproduction number), defined as the mean number of infections produced by a typical infectious case at calendar time $t$ [6]. A reliable estimation of $R_t$ is useful to quantify if the outbreak is declining ($R_t < 1$), growing ($R_t > 1$), or plateauing ($R_t = 1$), and can be used to evaluate and adjust health

interventions in real time [7]. A classical assumption in epidemic modeling is that in the early phase when susceptible depletion is negligible or control measures are absent, outbreaks show exponential growth dynamics. Nevertheless, early subexponential growth dynamics have already been documented in empirical data for multiple disease outbreaks including HIV and Ebola virus [8–10]. The mechanisms that lead to subexponential growth profiles are still debated but include heterogeneity in contact structures, clustering of contacts, and individual behavioral changes due to an increased risk perception [8]. These mechanisms have been present in the current outbreaks [1, 4]. Hence, we consider the possibility of early subexponential growth dynamics on the mpox outbreaks using a flexible phenomenological model that is able to reproduce several growth profiles [8].

The main goal of this work is to obtain a data-driven estimation of $R_t$ and growth rates for the mpox outbreak in the most affected European countries using incidence time series based on confirmed cases. To obtain a reliable estimation of the instantaneous reproduction number we use a recently proposed filtering type Bayesian inference [11]. This estimation considers the mean generation time for the current mpox outbreak. Furthermore, we pose a novel adaptive scheme to consider non-autonomous epidemiological dynamics, in particular, time-dependent dynamics of the intrinsic epidemic growth rate in a generalized growth model. To this end, the number of mpox cases is modeled through a sequential Bayesian approach [12] with a Negative Binomial (NB) model.

## Materials and methods

### The data

According to the European Center for Disease Control (ECDC), as of November 22, 2022, a total of 17,395 mpox cases have been confirmed in Europe. Spain (7,405), France (4,104), United Kingdom (3,720), Germany (3,672), the Netherlands (1,248), Portugal (948), and Italy (917) are the most affected countries in terms of confirmed cases within the European region [3]. We present daily numbers of laboratory-confirmed mpox cases from the start of the outbreak in each country, respectively, until 23 September 2022 (see Fig 1). Following this period most of these countries report very few cases for several weeks. To account for reporting delays

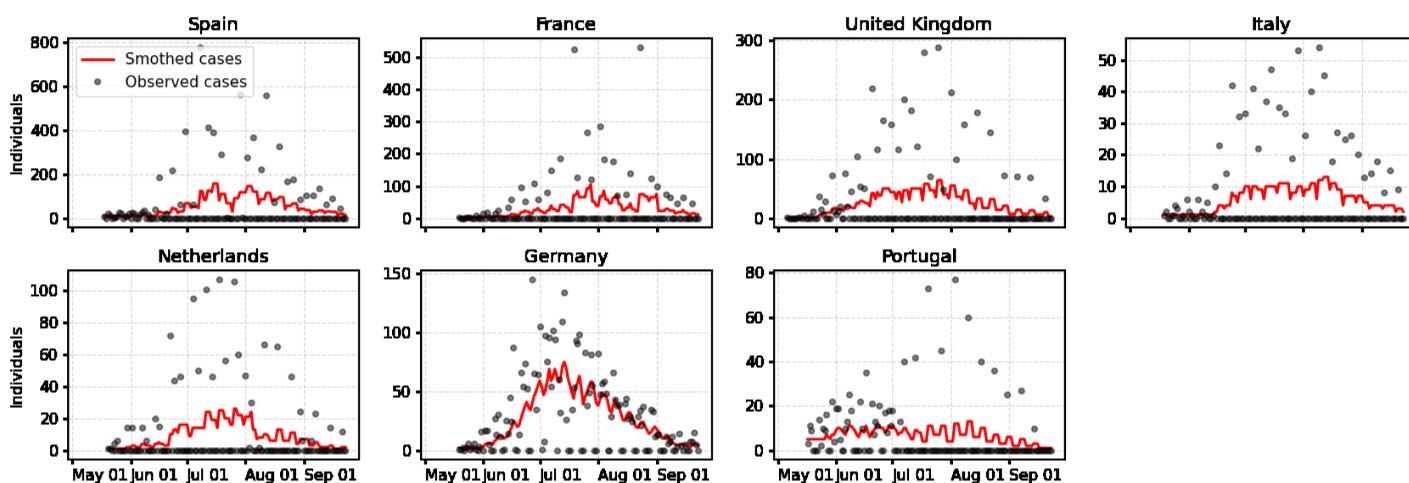

**Fig 1. Daily laboratory-confirmed mpox cases in Spain, France, the UK, Italy, the Netherlands, Germany, and Portugal since the beginning of the outbreak to September 23, 2022.** Raw incidence counts are presented as points and a 10-day moving average of (smoothed) cases is shown via a solid red line.

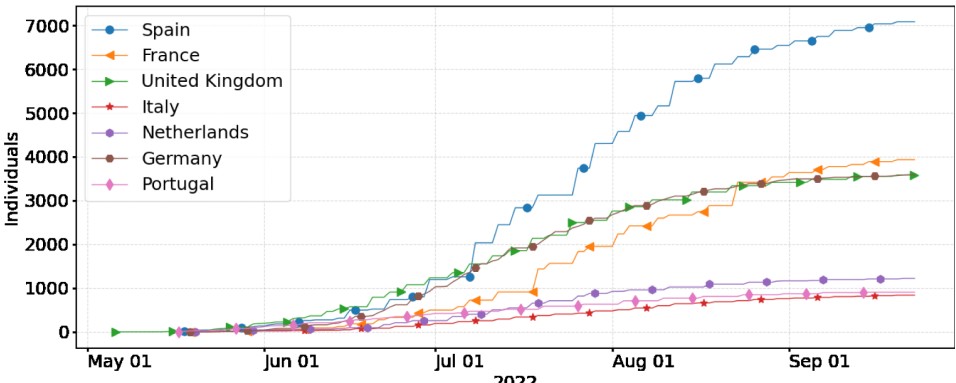

**Fig 2. The cumulative number of confirmed mpox cases in Spain, France, Germany, the UK, the Netherlands, Portugal, and Italy from the beginning of the outbreak to September 23, 2022.**

and other incidence data anomalies, a 10-day moving average of cases is also presented (see Fig 1). The incidence time series and confirmed cases for Europe and several other countries can be obtained from an open-access database presented in [13]. In the case of the European Region, data is also available via the European Center for Disease Control (ECDC) and the WHO Regional Office for Europe through The European Surveillance System (TESSy) [3]. Using the confirmed daily cases we obtained the cumulative cases in the European countries with the highest disease burden (see Fig 2).

## Basic and instantaneous reproduction numbers

The basic reproduction number, $R_0$, is a central concept in the modeling of infectious diseases. $R_0$ is defined as the expected number of secondary infections produced by a typical infected individual in a population where everyone is susceptible. Observe that $R_0$ considers a fully susceptible population and, hence, control measures or behavioral changes technically do not reduce its value. In the presence of time-dependent changes in the population, the instantaneous reproduction number, $R_t$, which is defined as the average number of secondary cases produced by one infected individual at time $t$ is a key metric for evaluating the transmissibility of infectious diseases. Both reproduction numbers have been used successfully to monitor contagion patterns in past epidemics including the COVID-19 pandemic [14, 15]. Several methods have been developed to estimate both reproduction numbers. For example, the basic reproduction number can be expressed as [16]

$$R_0 = \frac{1}{\int_0^\infty e^{-rt} g(t) dt}. \tag{1}$$

Here, the exponential growth rate, $r$, is defined as the per capita change in the number of new infections per unit of time. The function $g(t)$ is the density of the generation interval $G$ which is the time lag between infection in a primary case and a secondary case. The generation time distribution $g(t)$ should be estimated empirically by considering the time lag between confirmed infector-infectee pairs. Nevertheless, in practice, it is difficult to obtain and is commonly substituted with the serial interval distribution that measures the time from illness onset in the primary case to illness onset in the secondary case [17]. Several methods have been proposed to estimate $R_t$ using only incidence data e.g. [11, 17–19]. Using the renewal equation [19], the instantaneous reproduction number can be expressed in terms of the

incidence $I_t$ at time $t$, and the discretized probability distribution of the generation interval denoted $g_s$ as follows

$$R_t = \frac{I_t}{\sum_{s=1}^{t} g_s I_{t-s}} = \frac{I_t}{\Psi_t}. \tag{2}$$

The denominator $\Psi_t$ is usually interpreted as the total infectiousness of infected individuals [18]. Furthermore, $\Psi_t$ can also represent an estimation of the current number of active cases, so $R_t$ is the ratio of secondary cases produced by the actual total active cases [11]. Given the definition (2), Cori et al. [18] assumed that

$$\mathbb{E}(I_t | I_{t-1}, \dots, I_2, I_1) = R_t \Psi_t, \tag{3}$$

where $\mathbb{E}(X)$ denotes the expectation of a random variable $X$, and $I_t$ conditional on previous incidences follows a Poisson model i.e. $I_t | I_{t-1}, \dots, I_2, I_1 \sim Po(R_t \Psi_t)$. Nevertheless, over-dispersion is expected for contagious events, so a natural improvement on the Poisson model would be to consider a more general count data model such as a Negative Binomial. Here, we use a recently proposed method [11] to obtain a reliable estimation of $R_t$ that improves the Poisson sampling model by adding an autoregressive prior for the log of observed $R_t$'s. This results in a dynamic linear model which coupled with Bayesian updating forms a filtering type inference for the sequence of $log(R_t)$ (see the details in [11]).

One key input to calculate $R_t$ is the generation time distribution. The mean generation time for the current mpox outbreak in Italy [20] was estimated to be 12.5 days (95% CI of the mean: 7.5–17.3; 5th and 95th percentiles of the distribution 5–23 days). In particular, Guzzetta et al. found that the generation time follows a Gamma distribution with scale 2.57 and shape parameter 4.85 (see Fig 3). Here, it is assumed that this estimate for Italy is valid for the European region. Considering the possible presence of noise due to delays in reporting and other incidence data anomalies in epidemiological data, a common practice is to consider a smoothed version, $\mathbb{S}[I_t]$, of the incidence time series. Several methods are available to obtain $\mathbb{S}[I_t]$ curves

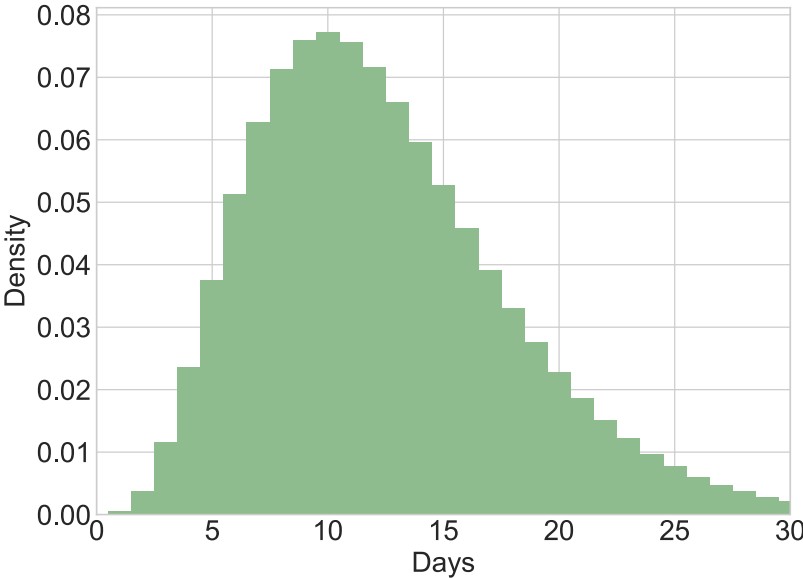

**Fig 3. Histogram for the estimated *Gamma*(4.85, 2.57) generation time distribution of the 2022 mpox outbreaks.**

e.g. splines of moving average filters (see [7] and the references therein). Since $R_t$ estimations based on very low incidence numbers are unreliable [18] and the mpox incidence time series present several days with zero cases, we consider the 10-day moving average for the daily cases presented in Fig 1.

## Phenomenological models

Phenomenological models have been proven to be useful tools to characterize and generate short-term forecasts of the evolution of epidemic outbreaks [9]. Phenomenological approaches prioritize the reproducibility of empirical observations. Hence, these models are particularly helpful in the presence of uncertainty of major epidemiological aspects including the potential contribution of multiple transmission pathways [8]. This is the case for mpox and other zoonotic diseases for which not only human-to-human infection is possible but also cross-species transmission via spillover events. Key quantities such as the growth rate, peak date, the epidemic's final size, and the epidemic wave's length can be estimated via phenomenological models. The early dynamics of epidemic outbreaks are usually characterized by exponential or sub-exponential growth, followed by a deceleration of growth due to control measures or behavioral changes in the population. Several phenomenological models such as the generalized growth model (GGM), the Richards model (RM), and the Blumberg hyper-logistic function have been used successfully to generate a variety of epidemic growth profiles observed in real epidemic outbreaks [8–10]. Most of these models can be obtained from the generalized logistic growth model (GLGM) [21] defined as

$$\frac{dC(t)}{dt} = rC(t)^p \left[1 - \left(\frac{C(t)}{K}\right)^a\right]^\delta \tag{4}$$

where $p$, $a$, and $\delta$ are non-negative real numbers.

Here, we consider the GGM model as a tool to characterize the epidemiological dynamics of the 2022 outbreak in the European region. The GGM is a particular case of the GLGM with $\delta = 0$, and is defined by the solution of the following differential equation:

$$\frac{dC(t)}{dt} = rC(t)^p, \tag{5}$$

where $C(t)$ represents the confirmed cumulative cases at time $t$, $r$ is the growth rate, and $p$ reflects the "deceleration of growth". The value of $p$ lies in the interval [0, 1] and allows the epidemic curve to mimic exponential ($p = 1$), sub-exponential ($0 < p < 1$), and linear growth ($p = 0$) [9]. The solution of the GGM can be easily obtained as

$$C(t) = [(1 - p)rt + C(0)^{1-p}]^{1/(1-p)}, \quad 0 < p < 1. \tag{6}$$

For $p = 1$, the solution is the classical exponential growth model $C(t) = C(0)e^{rt}$. Whereas for $p = 0$, we have the linear growth solution $C(t) = rt + C(0)$.

## Bayesian parameter inference

We perform a simultaneous estimation of parameters $r$ and $p$ that focus on the epidemic growth phase of the mpox outbreaks. To this end, we adopt a Bayesian statistical approach, which is well suited to model multiple sources of uncertainty and allows the incorporation of background knowledge on the model's parameters.

**Observational model and data.** The data used to fit our model is obtained by calculating the 10-day moving average of daily laboratory-confirmed cases, as illustrated in Fig 1. This

10-day smoothed daily time series is then used to represent daily cases in our model. The application of this smoothing technique addresses reporting delays and anomalies in the mpox incidence data, thereby improving the overall quality of the time series.

We defined the theoretical expectation $\mu$ of the smoothed daily cases at a discrete time $t_i$, as follow:

$$\mu(t_i) = C(t_i) - C(t_{i-1}),$$

where, $C(t_i)$ denotes the number of cases at time $t_i$, while $C(t)$ is determined by the subexponential solution (6) of the GGM model.

However, it is important to note that smoothing techniques alone do not adequately address the issue of overdispersion commonly observed in epidemiological data. Therefore, we account for this concern by utilizing a negative binomial (NB) distribution denoted as $NB$ $(\mu, \omega, \theta)$. This distribution incorporates parameters $\theta$ and $\omega$, which explicitly capture the over-dispersion phenomenon.

For the observed data points $y_i$, we assume the following distribution:

$$y_i \sim NB(\mu(t_i), \omega, \theta),$$

with fixed values for the over-dispersion parameters $\omega$, $\theta$. Conditional independence is assumed in the data and therefore, the likelihood is obtained from the NB model. The parameters to be inferred are the growth rate ($r$), the deceleration of growth ($p$), and we also infer the initial condition ($C_0$). To sample from the posterior, we employ the Markov Chain Monte Carlo (MCMC) method, specifically utilizing $t$-walk generic sampler [22]. The MCMC algorithm with the t-walk generic sampler runs semi-automatic. That is, while the MCMC algorithm performs most of the sampling procedure automatically, the user can adjust certain parameters based on their expertise and understanding of the problem. These adjustable parameters include the shape and parameters for the prior distributions, the likelihood function, and the initial conditions.

**Bayesian sequential approach.**  Modeling studies often consider epidemiological dynamics as an autonomous dynamical system and neglect time-dependent changes in epidemiological parameters [23]. Nevertheless, parameters usually evolve during the outbreak due to the impact of health interventions and changes in risk perception [24–26]. Here, we adapt the sequential data assimilation approach proposed in [12] to obtain time-varying parameters. This approach assumes that suitable transmission, epidemic, and observation models are available and coded into a dynamical system. The main idea is to train the model using only a subset of the most recent data. The estimation is updated sequentially in a sliding window of data. We apply this method to understand the disease dynamic in the growth stage of the outbreak and track changes over the estimated parameters in the phenomenological model (6) using incidence time series for mpox in European countries.

Regarding the elicitation of the parameters' prior distribution for the first forecast, we use a Gamma distribution for the initial condition, $C_0$, with scale 1 and shape parameter 10. This assumption is appropriate to model low incidence counts, near to 10, on the initial number of infected individuals. A Beta prior distribution is used for the deceleration of growth parameter $p$ as it is well-suited for modeling variables with values between 0 and 1. The initial parameters of the Beta distribution were chosen to provide a nearly uniform probability density across the range of values from 0 to 1. This selection allows for an unbiased and flexible prior assumption for the deceleration of growth.

**Table 1. Parameters and prior distributions for the first learning window used for Bayesian inference.**

| Parameter | Prior distribution |
|---|---|
| Growth rate $r$ | $Gamma(3, 1)$ |
| Deceleration of growth $p$ | $Beta(1 + 1/6, 1 + 1/3)$ |
| Initial condition $C_0$ | $Gamma(10, 1)$ |

The prior distributions presented here are only used at the start and are not used in the rest of the sequential inference, wherein each window, the prior is an over-dispersed version of the posterior in the previous window.

In modeling the growth rate ($r$), we assume a Gamma distribution that is particularly suitable for modeling positive quantities, which is relevant in the context of the outbreak being modeled in this case. For selecting parameters in the Gamma distribution, we prioritized assigning a higher probability to values between 0 and 10. This choice reflects our understanding that during the initial stages of an outbreak, the growth rate tends to be relatively higher. However, we also allocated probability to lower values to account for the possibility of lower growth rates occurring. For further details and a visual representation of the prior distributions, please refer to Table 1 and the S1 File.

We have to stress that prior distributions are only used at the first learning window. Then, the MCMC posterior sample from window $k$ is used to create a prior for the next window $k + 1$ (see details in [12]).

## Results

### The instantaneous reproduction number

The instantaneous reproduction number $R_t$ for the laboratory-confirmed mpox cases in Spain, France, Germany, the UK, the Netherlands, Portugal, and Italy from June 25 to September 23, 2022, is presented in Fig 4. We used a $Gamma(25, 0.5)$ distribution (see Fig 3) as an approximation of the generation time distribution for the European region. In Fig 4, the red solid line represents the median estimate for $R_t$, whereas the dark and light gray shaded areas represent 50% and 90% quantiles, respectively. The black solid line indicates the threshold value 1 on the reproduction number. At the beginning of the period, our estimation for $R_t$, which can be used as a proxy for the basic reproduction number $R_0$, was 2.06 (95% CI 1.63 − 2.54) for Spain, 2.62 (95% CI 2.23 − 3.17) for France, 2.81 (95% CI 2.51 − 3.09) for Germany, 1.82 (95% CI 1.52 − 2.18) for the UK, 2.84 (95% CI 2.07 − 3.91) for the Netherlands, 1.13 (95% CI 0.99 − 1.32) for Portugal, 3.06 (95% CI 2.48 − 3.62) for Italy.

Observe (see Fig 4) that although Italy exhibited the highest initial instantaneous reproduction number (considering cases from the last week of June), following this period their $R_t$ showed a clear decreasing trend during July reaching the threshold value 1 at the end of this month. The reproduction number for Germany showed similar patterns reaching its highest value in late June followed by a rapid decrease. The instantaneous reproduction number for Spain, France, and the UK also reached its highest value in late June. On the other hand, the $R_t$ for the Netherlands reached its highest value one week late, whereas the Portuguese $R_t$ showed oscillations around criticality during the whole time period. On average, the instantaneous reproduction numbers for Spain and France showed a value above one (supercritical dynamics) for a longer period of time. This is supported by the fact that both countries present the highest number of cumulative cases within the region. During the end of the period (September), all the countries considered here showed an instantaneous reproduction number below 1

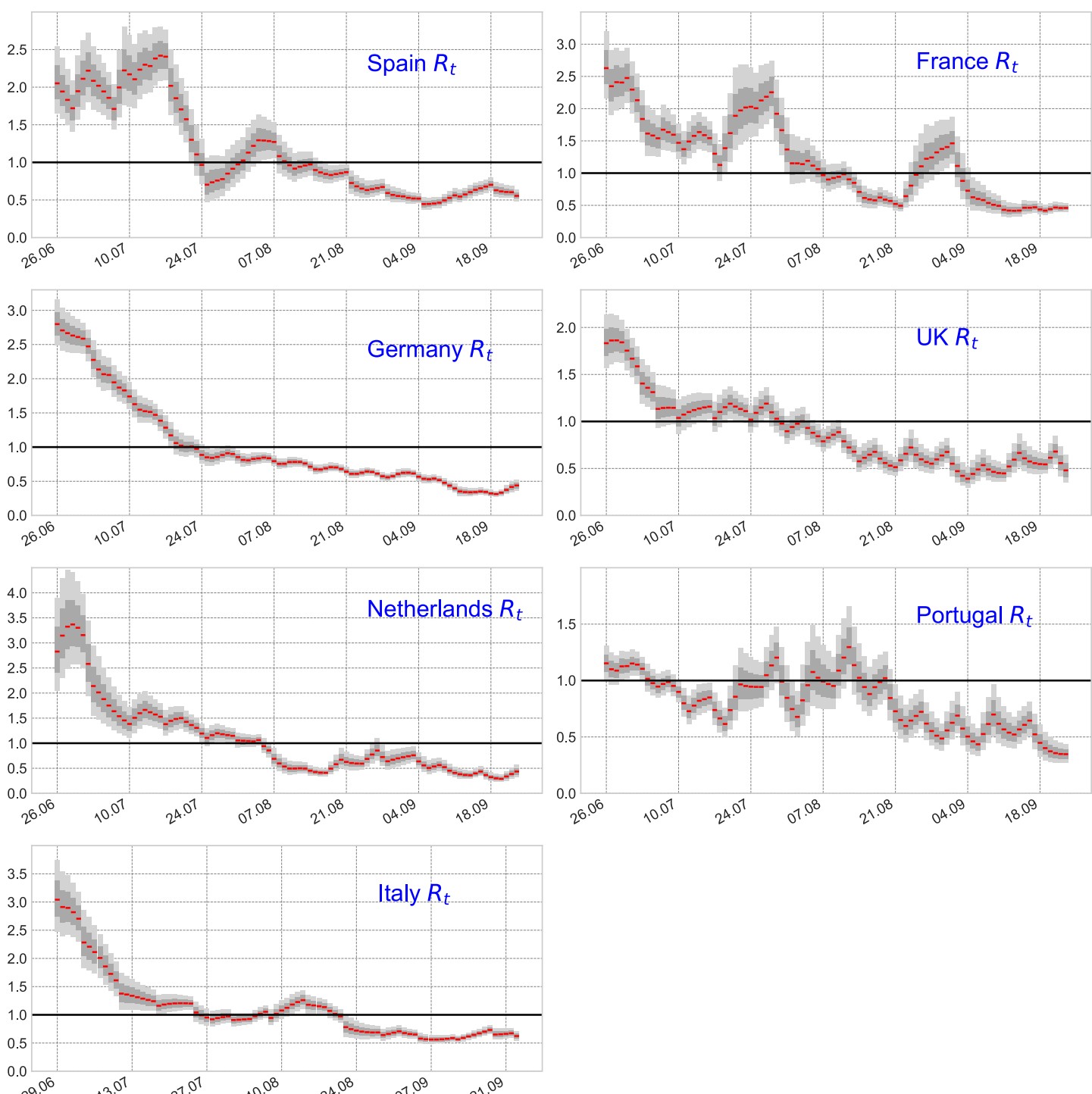

**Fig 4. The effective reproduction number $R_t$ for confirmed mpox cases in Spain, France, Germany, the UK, the Netherlands, Portugal, and Italy from June 25 to September 23, 2022.** The assumed generation time distribution for the computation of $R_t$ is a *Gamma*(4.85, 2.57) distribution. The red dots represent the median estimate for $R_t$. The dark and light gray shaded areas represent 50% and 90% quantiles for $R_t$, respectively. The black solid line indicates the threshold value 1 on the reproduction number.

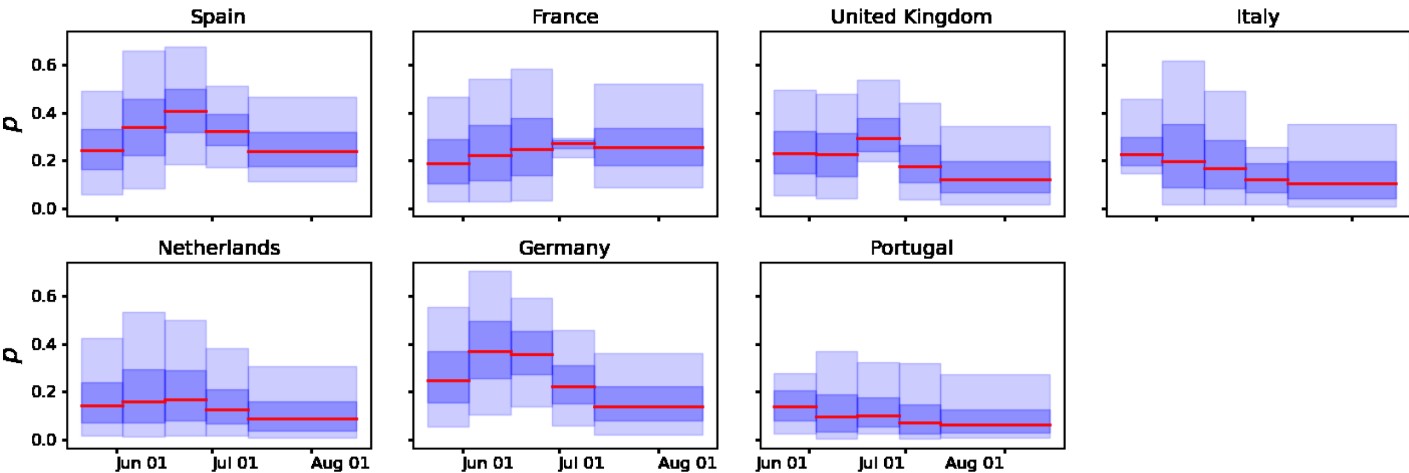

**Fig 5. Time-varying estimates of the deceleration of growth parameter, *p*, derived from fitting the solution of the GGM model (6) to the epidemic growth phase for confirmed daily mpox cases in Spain, France, Germany, the UK, the Netherlands, Portugal, and Italy, respectively.** The solid red line represents the median estimate. The dark and light blue shaded areas represent 50% and 90% quantiles, respectively.

which is in agreement with the significant reduction in daily confirmed mpox cases within the European area [3].

## Growth rates and subexponential dynamics

We performed the Bayesian sequential approach for parameter estimation in the GGM model (5) and its solution (6) to characterize the epidemic growth patterns of the outbreaks in the most affected European countries. The model is tested on the smoothed incidence time series presented in Fig 1. Figs 5 and 6 show the parameter estimation results for the parameters *p* and *r*, respectively. The red solid line represents the median estimate, whereas the dark and light blue shaded areas represent 50% and 90% quantiles, respectively. Fig 7

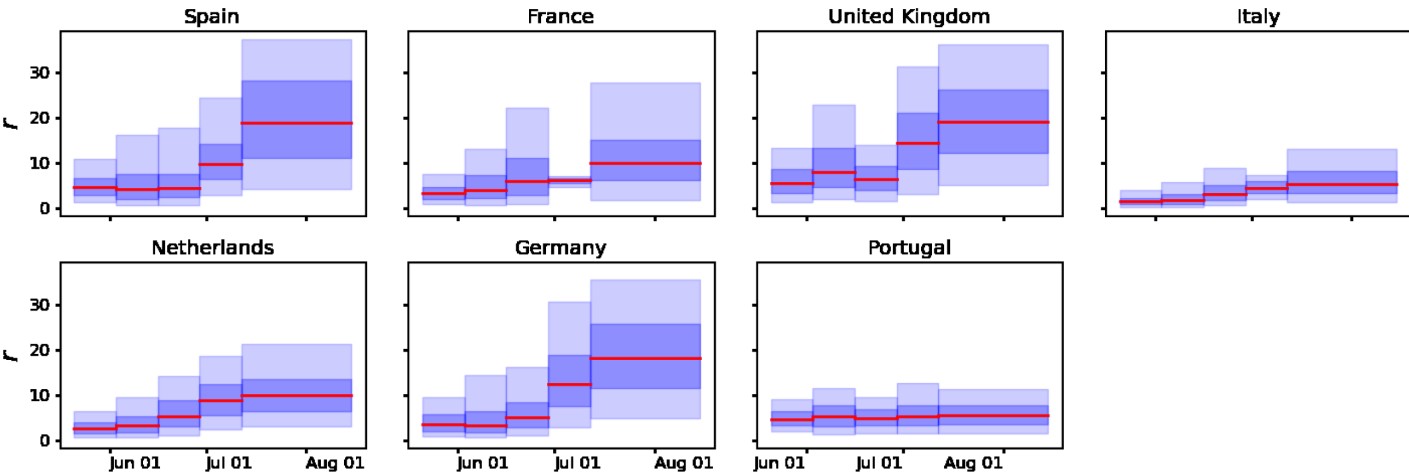

**Fig 6. Time-varying estimates of the growth rate, *r*, derived from fitting the solution of the GGM model (6) to the epidemic growth phase for confirmed daily mpox cases in Spain, France, Germany, the UK, the Netherlands, Portugal, and Italy, respectively.** The solid red line represents the median estimate. The dark and light blue shaded areas represent 50% and 90% quantiles, respectively.

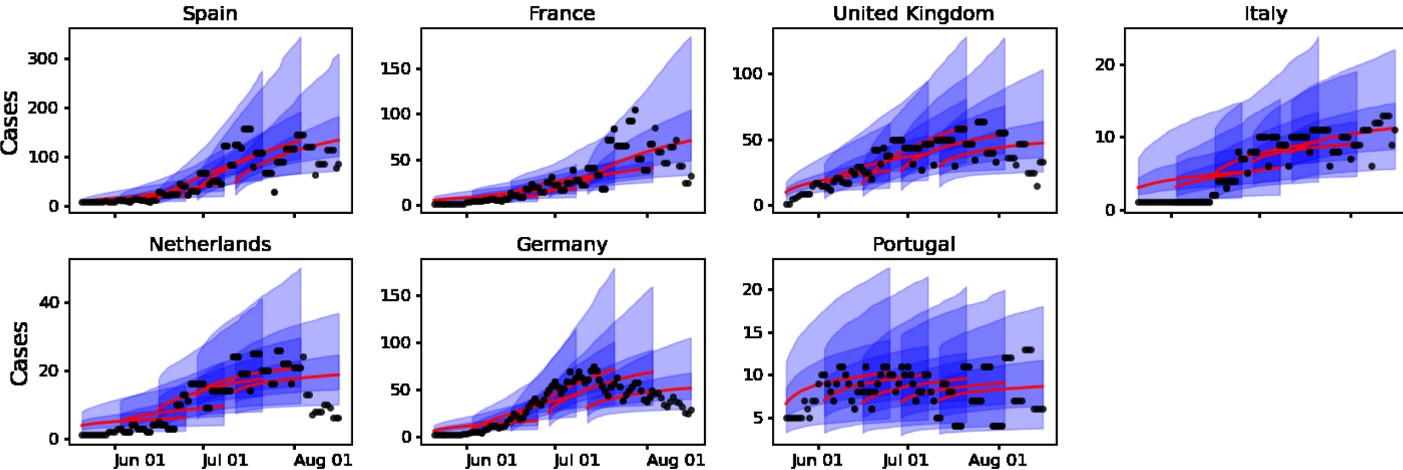

**Fig 7. Outbreak analysis for incidence time series of mpox cases in Spain, France, Germany, the UK, the Netherlands, Portugal, and Italy.** Solid red lines indicate the median incidence forecast. The darker-shaded blue region indicates the interquartile forecast range, and the lighter-shaded blue region indicates the 5–95th percentile range. Smoothed cases are presented as points.

presents the fits of the GGM model to the smoothed cases (see Fig 1). The estimation indicates that outbreaks in all the countries considered in this study (Spain, France, Germany, the UK, the Netherlands, Portugal, and Italy) show subexponential growth with most estimates of $p$ displaying mean values below 0.5 (see Fig 5). Current monkeypox outbreaks disproportionately affected the GBMSM community showing non-homogeneous mixing and clustering in contact patterns, mechanisms that have been linked to subexponential growth dynamics for several diseases [8]. The growth rate $r$ for most of the countries showed an average increasing pattern except for Portugal which showed an almost constant growth rate (see Fig 6). This coincides with the $R_t$ for Portugal whose value was oscillating around 1 during the whole outbreak (see Fig 4).

## Discussion

In this study, we have provided a reliable estimation of the instantaneous reproduction number $R_t$ for the mpox outbreaks in the European region during June—September 2022. The estimates reported here provide useful information about pathogen transmission and can assist with outbreak control. $R_t$ quantifies the potential for epidemic spread at time $t$ and the impact of intervention policies such as vaccination, contact tracing, and case isolation. In the case of supercritical dynamics ($R_t > 1$), the value of $R_t$ informs the proportion of new infections you need to prevent in order to achieve a stable or decreasing incidence [11, 18, 19]. We retrieved incidence time series on the daily laboratory-confirmed mpox cases available from an open-access database presented in [13]. We used a filtering type Bayesian inference that depends on a dynamic linear model on the log of observed $R_t$'s to compute the instantaneous reproduction number in the European countries with the highest burden of disease in terms of cumulative cases. We have considered a plausible generation time distribution based on a recent estimation of the mean generation time in Italy [20]. During the early phase of the outbreak, the instantaneous reproduction numbers in Spain, France, Germany, the UK, the Netherlands, Portugal, and Italy were all higher than 1 indicating supercritical epidemiological dynamics. Nevertheless, after late August most of these countries (except France) showed an $R_t$ value

below 1 which is in agreement with the observed reduction in mpox cases within the European area [3].

The current atypical mpox outbreaks have exposed important gaps in understanding the transmission patterns and continuously evolving epidemiological characteristics of the disease [4]. Hence, we considered a simple phenomenological model that can reproduce various epidemic growth profiles with no need for explicit mechanistic assumptions about the transmission dynamics [9]. To account for the non-autonomous nature of epidemic outbreaks and the time-dependent variables that influence them, we implemented an adaptive sequential Bayesian approach. This allowed us to estimate the model parameters in a time window, considering the dynamic nature of epidemic systems influenced by viral evolution, human behavior, prior immunity, and other time-dependent variables. This adaptive sequential Bayesian approach was initially presented in [12] for modeling the infection dynamics of COVID-19. The forecast performance of this approach was evaluated using prediction interval coverage, and the findings provided strong evidence of the model's ability to capture uncertainty and generate reliable predictions accurately. This adaptive sequential Bayesian approach has also been employed in other studies [27, 28] to capture the COVID-19 dynamics using different models and data types.

The results suggest that mpox outbreaks in all the countries considered in this study (Spain, France, Germany, the UK, the Netherlands, Portugal, and Italy) show subexponential growth with estimates of the deceleration of growth parameter, $p$, displaying mean values substantially lower than 1.0 (see Fig 5). Current mpox outbreaks disproportionately affected the GBMSM community showing non-homogeneous mixing and clustering in contact patterns, mechanisms that have been linked to subexponential growth dynamics in the past [8]. Our analysis is based on a Bayesian statistical approach that allows the incorporation of background knowledge in the model's parameters. Although our approach is well suited to model multiple sources of uncertainty, two major sources of error are worth to be mentioned. First, due to limited testing availability, especially at the early phase of the current outbreak, the daily laboratory-confirmed cases might be prone to underreporting. Second, the estimation of the instantaneous reproduction numbers is highly dependent on the generation time distribution, which has been only estimated for the case of Italy [20]. Furthermore, the models used in this study stress the reproducibility of empirical observations but neglect the biological mechanism behind the observed epidemiological dynamics [8–10].

In summary, in this study, we report the instantaneous reproduction numbers and time-dependent estimations of the epidemic growth rates at a country level in the European region. Our findings suggest that the recent mpox outbreaks resemble stuttering chains of transmission so the possibility of a huge outbreak is very low. These results agree with the recent predicted global decreasing trend in mpox cases found in [29] based on related growth models. Confirmed cases are decreasing significantly in the European region. Compared to the peak of the outbreak reached during the epidemiological week 29 (18–24 July 2022), in which 2, 151 cases were reported, there has been a reduction of more than 90% in the number of newly reported cases during week 42 (17–23, October 2022) [3]. This decreasing trend confirms the positive impact of the interventions to contain the outbreak in the affected regions. One of the primary measures was to increase education about the disease's symptoms and transmission. Then raise acceptance of preventive measures such as vaccination and limiting the number of sex partners to reduce the likelihood of exposure. The education intervention within the high-risk groups appeared to be a key factor driving the reduction in cases [29]. Other efforts to contain the outbreak include active case detection, contact tracing, and isolation of infected individuals [1, 2, 4, 29]. Although these measures have been successful to control the outbreak,

further strategies toward elimination are essential to avoid the subsequent evolution of the virus that can result in new outbreaks.

## Supporting information

**S1 File.**
(PDF)

## Author Contributions

**Conceptualization:** Fernando Saldaña, Maria L. Daza-Torres, Maíra Aguiar.

**Data curation:** Fernando Saldaña.

**Formal analysis:** Fernando Saldaña, Maria L. Daza-Torres.

**Funding acquisition:** Maíra Aguiar.

**Investigation:** Fernando Saldaña, Maria L. Daza-Torres.

**Methodology:** Fernando Saldaña, Maria L. Daza-Torres.

**Project administration:** Maíra Aguiar.

**Resources:** Maíra Aguiar.

**Software:** Fernando Saldaña, Maria L. Daza-Torres.

**Supervision:** Maíra Aguiar.

**Validation:** Maíra Aguiar.

**Visualization:** Fernando Saldaña, Maria L. Daza-Torres.

**Writing – original draft:** Fernando Saldaña, Maria L. Daza-Torres.

**Writing – review & editing:** Fernando Saldaña, Maria L. Daza-Torres, Maíra Aguiar.

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
