## [Decision Letter · Decision Letter 0]

5 Jun 2023

PONE-D-22-35033Data-driven estimation of the instantaneous reproduction number and growth rates for the 2022 monkeypox outbreak in EuropePLOS ONE

Dear Dr. Saldana,

Thank you for submitting your manuscript to PLOS ONE. After careful consideration, we feel that it has merit but does not fully meet PLOS ONE’s publication criteria as it currently stands. Therefore, we invite you to submit a revised version of the manuscript that addresses the points raised during the review process.

We look forward to receiving your revised manuscript.

Kind regards,

Julian Ruiz-Saenz

Academic Editor

PLOS ONE

Journal Requirements:

“FS and MA are supported by the Basque Government through the ``Mathematical Modeling Applied to Health'' Project, BERC 2022-2025 program and by the Spanish Ministry of Sciences, Innovation and Universities: BCAM Severo Ochoa accreditation SEV-2017-0718. The funders had no role in study design, data collection and analysis, decision to publish, or preparation of the manuscript.”

“This research is supported by the Basque Government through the “Mathematical Modeling Applied 289 to Health” Project, BERC 2022-2025 program and by the Spanish Ministry of Sciences, Innovation 290 and Universities: BCAM Severo Ochoa accreditation SEV-2017-0718”

“FS and MA are supported by the Basque Government through the ``Mathematical Modeling Applied to Health'' Project, BERC 2022-2025 program and by the Spanish Ministry of Sciences, Innovation and Universities: BCAM Severo Ochoa accreditation SEV-2017-0718. The funders had no role in study design, data collection and analysis, decision to publish, or preparation of the manuscript.”

Reviewers' comments:

Reviewer's Responses to Questions

**Comments to the Author**

1. Is the manuscript technically sound, and do the data support the conclusions?

Reviewer #1: Yes

Reviewer #2: Yes

2. Has the statistical analysis been performed appropriately and rigorously? 

Reviewer #1: Yes

Reviewer #2: Yes

3. Have the authors made all data underlying the findings in their manuscript fully available?

Reviewer #1: Yes

Reviewer #2: Yes

4. Is the manuscript presented in an intelligible fashion and written in standard English?

Reviewer #1: Yes

Reviewer #2: Yes

5. Review Comments to the Author

Reviewer #1: This is an interesting study reporting estimates of reproduction number of the monkeypox epidemics in several European countries. I enjoyed reading the paper, and only have a few clarifying comments:

1)Did authors employ the 10-day smoothed daily time series or the raw time series? Initially I thought they had employed the smoothed series presented in figure 1. However, authors accounted for the overdispersion in the observation model when fitted the models to the data. Is it necessary to characterize overdispersion if the smoothed time series are used?

2)Authors employed a Gamma (25, 0.5) distribution to model the generation time distribution of monkeypox. Did authors obtain this distribution directly from the Gozzueta et al EID article? Just to be sure that authors are using the same distribution inferred in the Gozzueta paper.

3)Authors may want to explore the role of using aggregated weekly data which may help stabilize the variability in the data series.

4) The Bayesian sequential approach employed to characterize the temporal uncertainty of the parameters looks interesting. I would like authors to confirm that the methodology works well with simulated data. Perhaps authors could generate a simulated data from the GGM with one step change in “r” and show that the Bayesian approach is useful to infer the correct parameter estimates. How much data is needed to correctly characterize the temporal changes in the parameters?

5)Although the goal of the paper is not forecasting the monkeypox epidemics, authors may find interesting that another paper has used related growth models to generate short-term forecasts of the monkeypox epidemics in various areas of the world. The models have yielded competitive forecasts . see: https://bmcmedicine.biomedcentral.com/articles/10.1186/s12916-022-02725-2.

Reviewer #2: The authors of this article focused on estimating the instantaneous reproduction number and growth rates for monkeypox using confirmed cases in Europe. They used a filtering type Bayesian inference to estimate the instantaneous reproduction number and a generalized growth model approximate the epidemic growth rate. The key takeaway from this paper is that a large outbreak of monkey box is low as current propagation of the disease bears a resemblance to stuttering transmission chains. Overall, the article is well-written with insightful results, and the authors make the dissemination of the results easy for the reader to follow.

Comments:

1)Page 2, Line 93. 1248 should be 1,248.

2)Page 4, Line 122. Citations should be in ascending order.

3)Page 4, Equation 2. I would recommend changing Lambda_t to another parameter as it is typically reserved for birth/recruitment rates.

4)Page 6, Line 182. The t in t-walk should be in mathmode.

5)Page 6, Line 182. “The MCMC runs semi-automatic,…” Please clarify this sentence.

6)Page 6, Table 1. What is the intuition for using a Beta prior distribution for the deceleration of growth, p?

7)Page 6-7, Section 3.1 and 3.2. The authors provide a nice summary of results with respect to the instantaneous reproduction number, growth rates, and subexponential dynamics. I think it would be beneficial to this section if there was some biological discussion on what is influencing these results, e.g. human behavior, disease characteristics, etc.

8)Page 7-8, Section 4. Can the authors highlight the robustness of this method? E.g. even though the focus was on monkeypox, how flexible is this methodology to other infectious diseases?

6. PLOS authors have the option to publish the peer review history of their article (what does this mean?). If published, this will include your full peer review and any attached files.

Reviewer #1: No

Reviewer #2: No

---

## [Author Response · Author response to Decision Letter 0]

20 Jul 2023

Please see the attachment with the response.

---

## [Decision Letter · Decision Letter 1]

8 Aug 2023

Data-driven estimation of the instantaneous reproduction number and growth rates for the 2022 monkeypox outbreak in Europe

PONE-D-22-35033R1

Dear Dr. Saldana,

We’re pleased to inform you that your manuscript has been judged scientifically suitable for publication and will be formally accepted for publication once it meets all outstanding technical requirements.

Kind regards,

Julian Ruiz-Saenz

Academic Editor

PLOS ONE

Additional Editor Comments (optional):

Reviewers' comments:

Reviewer's Responses to Questions

**Comments to the Author**

1. If the authors have adequately addressed your comments raised in a previous round of review and you feel that this manuscript is now acceptable for publication, you may indicate that here to bypass the “Comments to the Author” section, enter your conflict of interest statement in the “Confidential to Editor” section, and submit your "Accept" recommendation.

Reviewer #1: All comments have been addressed

Reviewer #2: All comments have been addressed

2. Is the manuscript technically sound, and do the data support the conclusions?

Reviewer #1: Yes

Reviewer #2: Yes

3. Has the statistical analysis been performed appropriately and rigorously? 

Reviewer #1: Yes

Reviewer #2: Yes

4. Have the authors made all data underlying the findings in their manuscript fully available?

Reviewer #1: Yes

Reviewer #2: Yes

5. Is the manuscript presented in an intelligible fashion and written in standard English?

Reviewer #1: Yes

Reviewer #2: Yes

6. Review Comments to the Author

Reviewer #1: (No Response)

Reviewer #2: (No Response)

7. PLOS authors have the option to publish the peer review history of their article (what does this mean?). If published, this will include your full peer review and any attached files.

Reviewer #1: No

Reviewer #2: No

---

## [Editor Report · Acceptance letter]

22 Aug 2023

PONE-D-22-35033R1 

Data-driven estimation of the instantaneous reproduction number and growth rates for the 2022 monkeypox outbreak in Europe 

Dear Dr. Saldaña:

I'm pleased to inform you that your manuscript has been deemed suitable for publication in PLOS ONE. Congratulations! Your manuscript is now with our production department. 

Kind regards, 

on behalf of

Dr. Julian Ruiz-Saenz 

Academic Editor

PLOS ONE